

# Positive peritoneal lavage fluid cytology based on isolation by size of epithelial tumor cells indicates a high risk of peritoneal metastasis

Ying Xuan[1,2,*], Qizhong Gao[2,*], Chenhu Wang[2] and Dongyan Cai[2]

[1] Jiangnan University, Wuxi School of Medicine, Wuxi, China
[2] Affiliated Hospital of Jiangnan University, Department of Oncology, Wuxi, China
* These authors contributed equally to this work.

## ABSTRACT

**Background:** Peritoneal metastasis (PM) is the most prevalent type of metastasis in patients with gastric cancer (GC) and has an extremely poor prognosis. The detection of free cancer cells (FCCs) in the peritoneal cavity has been demonstrated to be one of the worst prognostic factors for GC. However, there is a lack of sensitive detection methods for FCCs in the peritoneal cavity. This study aimed to use a new peritoneal lavage fluid cytology examination to detect FCCs in patients with GC, and to explore its clinical significance on diagnosing of occult peritoneal metastasis (OPM) and prognosis.

**Methods:** Peritoneal lavage fluid from 50 patients with GC was obtained and processed *via* the isolation by size of epithelial tumor cells (ISET) method. Immunofluorescence and fluorescence *in situ* hybridization (FISH) were used to identify FCCs expressing chromosome 8 (CEP8), chromosome 17 (CEP17), and epithelial cell adhesion molecule (EpCAM).

**Results:** Using a combination of the ISET platform and immunofluorescence-FISH, the detection of FCCs was higher than that by light microscopy (24.0% *vs.* 2.0%). Samples were categorized into positive and negative groups, based on the expressions of CEP8, CEP17, and EpCAM. Statistically significant relationships were demonstrated between age ($P = 0.029$), sex ($P = 0.002$), lymphatic invasion ($P = 0.001$), pTNM stage ($P = 0.001$), and positivity for FCCs. After adjusting for covariates, patients with positive FCCs had lower progression-free survival than patients with negative FCCs.

**Conclusion:** The ISET platform highly enriched nucleated cells from peritoneal lavage fluid, and indicators comprising EpCAM, CEP8, and CEP17 confirmed the diagnosis of FCCs. As a potential detection method, it offers an opportunity for early intervention of OPM and an extension of patient survival.

Corresponding author
Dongyan Cai, doctorcai@aliyun.com

# INTRODUCTION

Gastric cancer (GC) is one of the most common malignant tumor types and the fifth leading cause of cancer-related mortality worldwide (*Bray et al., 2024*). Although radical surgery is the standard treatment, the 5-year survival rate is 20–40%, with China reporting 35.1% (34.5 to 35.7) in 2012–2015 (*Zeng et al., 2018*). Late-stage diagnoses, particularly when there is serosal invasion, lead to treatment failure, resulting in nearly half of the patients succumbing to peritoneal metastasis (PM) (*Chen et al., 2021*).

PM is a frequent metastasis pattern in GC and remains a challenging clinical issue. In the era of precision therapy, ongoing research focuses on molecular characterization and biomarkers to predict PM (*Chen et al., 2020*; *Kolomanska & Gluszek, 2020*). Diverse prediction models, including PET-CT radiomics features (*Xie et al., 2023*) and those for late-stage GC occult PM (*Gao et al., 2023*), are being explored. Emerging technologies like salivary cell-free DNA detection (*Swarup et al., 2023*) are being investigated, alongside established methods like SE-iFISH for PD-L1 expression and the karyotype of circulating tumor cells (CTCs)/circulating endothelial cells (CECs) (*Chong et al., 2023*). In addition, ctDNA-based molecular residual disease (MRD) detection (*Wang, Li & Zhao, 2023*), is also contributing to the prediction of PM in GC. Simultaneously, numerous studies have investigated the molecular mechanisms of PM in GC. Galectin-1, Transgelin-2(TAGLN2), and others have been found to contribute to PM (*Shen et al., 2023*; *Ji et al., 2023*), while troponin I2, cellular retinoic acid binding protein 1(CRABP1), and others serve as predictors (*Sawaki et al., 2018*; *Sakata et al., 2022*).

The aforementioned studies have advanced our understanding of PM and have been a guide to its detection. Histopathological examination of peritoneal tissue biopsy is the gold standard for identifying PM, but it only detects macroscopic metastases. Without suspicious nodules, diagnostic laparoscopy combined with peritoneal lavage fluid cytology examination is a reasonable and standard method for diagnosing occult peritoneal metastasis (OPM) invisible to the naked eye. A prospective study showed that using laparoscopic exploration and cytology, 47% of patients changed their initial treatment strategy due to the discovery of OPM (*Irino et al., 2018*). The presence of free peritoneal cancer cells is a prerequisite for PM and is an independent poor prognostic factor for GC (*Lee et al., 2012*), basing on the theory that free cancer cells (FCCs) in the abdominal cavity act as seeds for the development of PM (*Dong et al., 2019*). Owing to the scanty number of cancer cells among the shed cells, detection methods often yield higher false-negative rates. Furthermore, clinical diagnosis is subjective and lacks standardization. Traditional peritoneal lavage cytology relies entirely on cell morphology, demonstrating a sensitivity of less than 60% for diagnosing OPM (*Hoskovec et al., 2017*). Thus, developing effective means to isolate and detect these cells in the abdominal cavity is crucial (*Wong & Coit, 2012*; *Taffon et al., 2019*). Progress in ascitic fluid cytology is limited; ongoing efforts involve microfluidic chip-based approaches (*Zhao et al., 2023*), optically induced electrokinetics (OEK) (*Zhang et al., 2020*) for high-throughput ascites and FCCs detection, and AI-assisted stimulated Raman molecular cytology (SRMC) for precise and swift detection of PM in GC (*Chen et al., 2023*). SRMC, a chemical microscopy-based intelligent

cytology, achieved 81.5% sensitivity and 84.9% specificity. A label-free OEK microfluidic technique was introduced for efficient isolation of GC cells from the ascitic fluid of six patients, achieving a remarkable purity of 71%. Yet, the clinical effectiveness of these methods awaits validation in terms of patient prognosis, posing a current challenge.

Currently, physical methods for cell separation mainly include microfluidic and the isolation by size of epithelial tumor cells (ISET) technology. Both of them are high-throughput technologies for isolating single cells and preparing them for single-cell sequencing, enabling the analysis of tumor cell heterogeneity (*Kang et al., 2019*). Microfluidic equipment is complex and costly, with sorting results influenced by gravity, buoyancy and fluid resistance, limiting its widespread clinical application. ISET platform, a cell size-based membrane filtration method, efficiently enriches CTCs, and was able to identify them in 100% of patients with hepatocellular carcinoma (*Morris et al., 2014*). Out-performing current immunology-based methods (*Yang et al., 2021*; *Hoshino et al., 2017*; *Mori et al., 2000*), ISET offers high retention, enrichment rates, and sensitivity, providing user-friendly, time-efficient, and accurate results. Importantly, the enriched and separated cells retain viability for subsequent experiments, such as molecular pathology detection through immunofluorescence and fluorescence *in situ* hybridization (FISH). Furthermore, after extracting the DNA, these cells can undergo genetic analysis, including hotspot mutation detection or single-cell sequencing. Selected IF-FISH biomarkers must meet key criteria: they must be cancer-specific, be able to distinguish cancer from normal cells, and be sensitive enough to detect trace amounts of cancer. Additionally, extensive research is needed to establish their strong correlation with clinical characteristics and tumor progression.

Chromosomal instability and aneuploidy are common in human malignancies (*Dyson, 2016*). Studies reveal specific chromosomal polyploidy in GC, such as CEP8 and CEP17 (*Costa Guimarães et al., 2006*; *Ciesielski et al., 2015*). The epithelial cell adhesion molecule (EpCAM or CD326) is highly expressed in various epithelial tumors, including esophageal, gastric, colorectal, lung, and ovarian cancer, with nearly 100% expression (*Eslami, Cortes-Hernandez & Alix-Panabieres, 2020*; *Hoogstins et al., 2017*). EpCAM overexpression is considered an early indicator of certain malignant tumors or precancerous lesions (*Zhang et al., 2017*; *Gires et al., 2020*) and associated with tumor size and lymph node metastasis in patients with GC. Moreover, the 5-year overall survival in patients with GC overexpressing EpCAM was lower than that in EpCAM-negative patients (*Dai et al., 2017*).

Utilizing ISET, we extracted free epithelial cells from GC patients. Analysis *via* FISH (CEP8 and CEP17 probes) and immunofluorescence (EpCAM) detected aneuploid and tumor cells. We then assessed FCCs for OPM and prognosis prediction in GC.

# MATERIALS AND METHODS

## Study design and participants

From August 2019 to August 2022, 50 patients diagnosed with GC without PM who required surgery were enrolled into the study at the Affiliated Hospital of Jiangnan University. The inclusion criteria were histological diagnosis of carcinoma before surgery; absence of distant organ metastasis, based on imaging examinations; lack of surgical

contraindications; ability to tolerate laparoscopic exploratory surgery; available clinicopathological data; and completion of a 3-year postoperative follow-up. Patients were excluded if they had a history of multiple tumors; uncontrollable infections; severe cardiovascular and cerebrovascular diseases; impaired function of the liver, kidney, or other organs; or were pregnant and breastfeeding. The study was conducted in accordance with the Declaration of Helsinki, and approved by the Ethics Committee of the Affiliated Hospital of Jiangnan University (approval number: LS2020060 on December 20, 2020). All participants provided written informed consent, following the ethical guidelines of the World Medical Association, Declaration of Helsinki for human research.

## Workflow

Laparoscopic exploration was performed on 50 patients with GC diagnosed without PM by preoperative imaging. Perform peritoneal nodule biopsy if any suspicious nodules were seen. Peritoneal lavage fluid (PLF) cytology was carried out in the absence of suspicious nodules. The collected PLF was subjected to traditional cytology and ISET+IF+FISH examination (Fig. 1).

## Laparoscopic exploration

The sequence of laparoscopic exploration was as follows: the left and right diaphragm, parietal peritoneum of the liver and spleen, pelvic cavity, greater omentum, small intestine, mesentery, transverse colon mesentery, and local lesions (*Li & Ji, 2015*; *Ding et al., 2020*). The following parameters were evaluated: presence of ascites; metastasis on the liver surface; peritoneal, mesentery, and omental metastatic lesions; the condition of the Douglas pouch; swollen gastric lymph nodes; and infiltration of the gastric serosal surface. Thereafter, the stomach wall was checked for stiffness and highly suspicious tissue regions were biopsied for pathological examination, to establish a definitive diagnosis. Patients with definitive organ metastases or PM detected at preoperative staging were excluded.

## Obtaining PLF

Approximately 250 ml of sterile saline was instilled into the left and right diaphragmatic roof, abdominal and pelvic parietal peritoneum, small intestine, and transverse mesocolon. Direct washing of any local lesions was avoided. In a reverse Trendelenburg position, a suction device was used to collect the flushing fluid from the pouch of Douglas, subhepatic, and splenic fossae. This was immediately sent to the pathology department (*Li & Ji, 2015*).

## Cytopathology

The amount of PLF collected from all patients was roughly the same. A low-speed desktop centrifuge was used to centrifuge PLF at 2,000 rpm for 10 min. The supernatant was removed, whereas the precipitate was spread onto two clean glass slides and fixed in ethanol (95%) for 15 min. After staining with Papanicolaou method, it was observed and diagnosed under a light microscope by the same pathologist and reviewed by another one. Atypical cytology was characterized by the following findings: nuclear irregularity, ratio of nucleus to cytoplasm (N/C) $\geq 0.8$, increased nuclear chromatin, anisonucleosis (nuclear size difference $\geq 1.5$), signet ring cells (characteristic ring-shaped nuclei due to

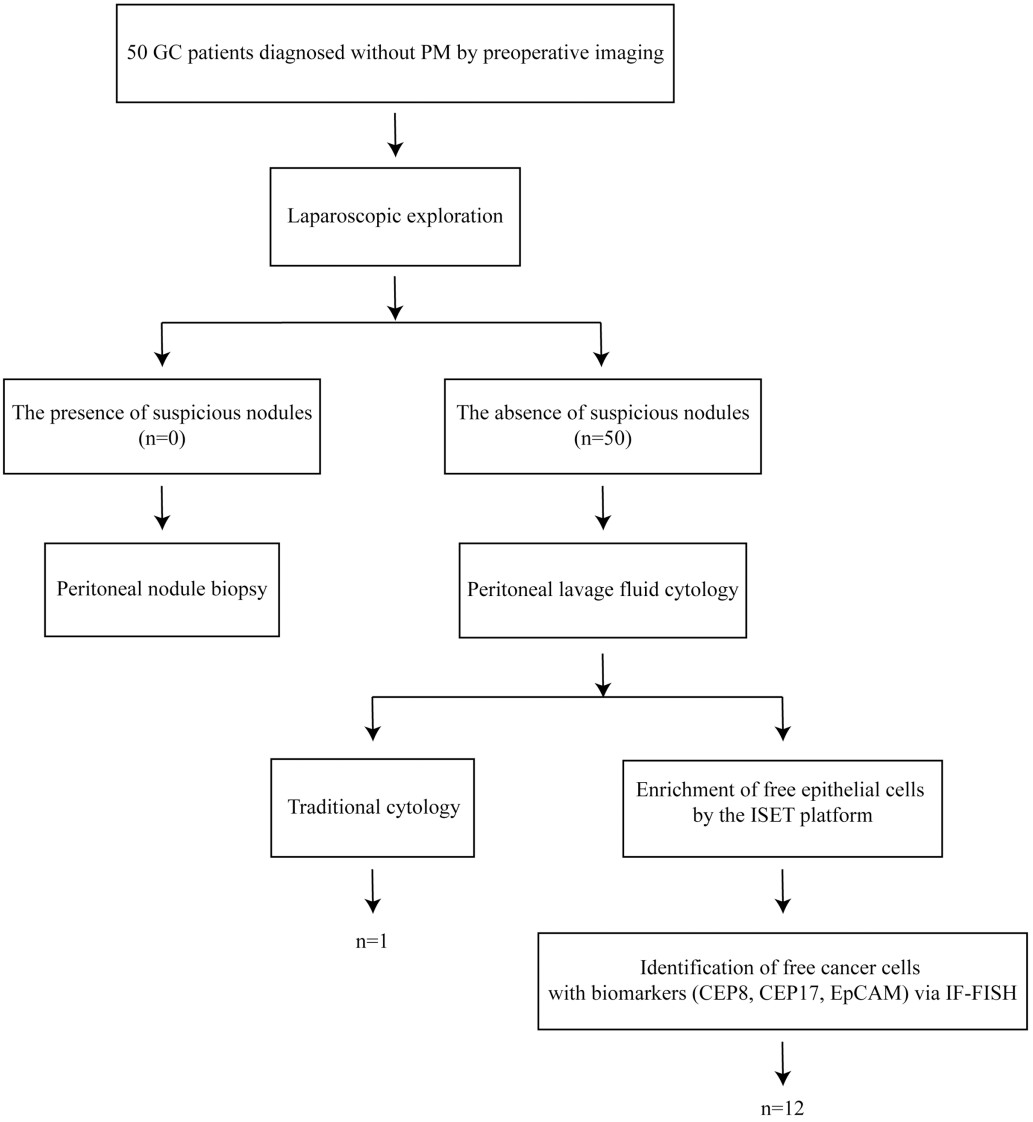

**Figure 1 The workflow of this study.** Laparoscopic exploration was performed on 50 patients with GC diagnosed without PM by preoperative imaging. Perform peritoneal nodule biopsy if any suspicious nodules were seen. Peritoneal lavage fluid (PLF) cytology was carried out in the absence of suspicious nodules. The collected PLF was subjected to traditional cytology and ISET+IF+FISH examination.

intra-cytoplasmic mucin), intracytoplasmic lumen, and cell cluster formation (three-dimensional aggregation of ≥3 cancer cells) (*Higaki et al., 2017*).

## Enrichment of free epithelial cells in PLF by the ISET platform

Abnormal cells were enriched from PLF by the ISET platform with the CTCBIOPSY A10 device (Wuhan YZY Medical Science and Technology Co., Ltd., Wuhan, China). The process was in accordance with the manufacturer's instructions, and the enriched cells was collected on a filter membrane that was fabricated *via* CTCBIOPSY patented technology (Wuhan YZY Medical Science and Technology Co., Ltd., Wuhan, China).

## Fluorescence *in situ* hybridization and immunofluorescence

The obtained cells were fixed on glass slides with 100 μl of 4% paraformaldehyde and dried at 33 °C. The CEP8 and CEP17 probes (Wuhan YZY Medical Science and Technology Co., Ltd.,Wuhan, China) were then added. After mounting, the specimen was hybridized using an automatic hybridization instrument (SH2000, Hangzhou Ruicheng Instrument Co., Ltd., Hanghzou, China) for 1.5 h. The cells were then incubated with 200 μl antibody preparation solution-1, 1 μl anti-EpCAM polyclonal antibody conjugated to Alexa Fluor 546, and 1 μl anti-hematopoietic white blood cell marker (CD45) polyclonal antibody conjugated to Alexa Fluor Plus 488 (Thermo Fisher Scientific, Waltham, MA, USA). The slides were then incubated for 2 h in a dark and humidified chamber. The nuclear dye 4′, 6-diamidino-2-phenylindole (DAPI, Thermo Fisher Scientific,Waltham, MA, USA) was finally added, and slides were mounted for examination with an Olympus BX53 fluorescent microscope (Olympus, Tokyo, Japan) and Pathfinder FISHIMSTAR automatic FISH scanning analysis system software (Thermo Fisher Scientific, Waltham, MA, USA).

## Interpretation of FCCs results

The enriched cells were meticulously examined for CEP8, CEP17, EpCAM, and CD45 markers. CD45-positive cells were identified as leukocytes. A cell was deemed a tumor cell if it exhibited three or more fluorescent signals for CEP8/CEP17 in the nucleus. Moreover, EpCAM expression in the cell membrane was indicative of a tumor cell. A positive determination of FCCs was made if even a single tumor cell was identified. Cells exhibiting CD45-CEP8/17+EpCAM+, CD45-CEP8/17-EpCAM+, and CD45-CEP8/17+EpCAM- were considered cancer cells. Otherwise, it was classified as negative. This rigorous examination ensured a comprehensive evaluation of the enriched cells based on multiple markers, contributing to the precision of the analysis.

## Treatment and follow-up

All patients were treated with oxaliplatin and capecitabine or S-1 adjuvant chemotherapy for 6–8 cycles after the operation. Full abdominal and chest CT-enhanced scans were performed mid-treatment, and after treatment completion. An evaluation was then performed according to the Response Evaluation Criteria in Solid Tumors 1.1. For patients with no recurrence or metastasis, the original treatment was continued; otherwise, follow-up was terminated.

## Statistical analysis

SPSS 26.0 software (IBM Corp., Armonk, NY, USA) was used for data analysis. Count data were expressed as cases (%), and clinicopathological variables were analyzed by the chi-squared ($\chi 2$) test or Fisher's exact test. GraphPad Prism 9.5.0 (GraphPad Software, San Diego, CA, USA) was used to generate Kaplan-Meier survival curves, and the log-rank test was used to assess the difference in progression-free survival (PFS). Multivariate Cox regression models incorporated with potentially influencing variables were used to calculate hazard ratios and 95% confidence intervals. The clinical variables in these models were chosen to address potential confounding factors. The receiver operating

characteristics (ROC) curve was plotted to assess the model's sensitivity and specificity. All tests were two-sided, and statistical significance was set at $P < 0.05$.

## RESULTS

### Clinical characteristics of the patients

A total of 50 patients with GC were enrolled in the present study, with a median age of 68 and a median follow-up time of 30.93 months. In addition, the sex ratio of females to males was approximately 3.5:1. The proportions of patients in stages I, II, III, and IV were 12.0% (6/50), 24.0% (12/50), 60.0% (30/50), and 4.0% (2/50), respectively.

### Detection of FCCs in the peritoneal cavity

The percentage of patients categorized as EpCAM+CEP8/17−, EpCAM-CEP8/17+, and EpCAM+CEP8/17+ was 4.0% (2/50), 18.0% (9/50), and 2.0% (1/50), respectively. Figure 2A shows that GC cells were positive for CEP8/17 and EpCAM. Cases were considered as positive if they exhibited at least three CEP8 or CEP17 signals in the cell nucleus (*Gao et al., 2016*) or EpCAM expression in the cell membrane (Fig. 2B). Among the enrolled patients, patient a was EpCAM positive and CEP8/17 negative, patient b was EpCAM negative and CEP8/17 positive, and patient c was EpCAM and CEP8/17 positive. Under a light microscope, cancer cells appeared to form cell clusters (Fig. 2C). In individuals with GC, the detection rate *via* ISET+IF+FISH was 24.0% (12/50), compared with 2.0% (1/50) by light microscopy.

### Relationships between clinicopathological factors and FCCs

Statistically significant relationships were demonstrated between age ($P = 0.029$), sex ($P = 0.002$), lymphatic invasion ($P = 0.001$), pTNM stage ($P = 0.001$), and positivity for FCCs (Table 1). Patients aged 68 years and above, of female sex, with lymph node invasion, and stage III disease or above, were more likely to have PLF that was positive for FCCs. The positivity for FCCs did not show any statistically significant relationships with depth of invasion, or venous and nerve invasion.

### Correlation between FCCs and survival

A 3-year follow-up was conducted, during which, with the exception of one case, the proportions of patients with PM, liver metastasis, and lung metastasis were 69.2% (9/13), 15.4% (2/13), and 15.4% (2/13), respectively. Moreover, two patients with PM experienced liver or lung metastasis. The median PFS for stages I–II, IIIA–IIIB, and IIIC–IV was 31.22, 31.45, and 11.27 months, respectively (Table S1). In all patients with validated PM, the ISET+IF+FISH detection rate was 33.3% (3/9) compared with 11.1% (1/9) for traditional cytology. Lymphatic invasion, pTNM stage, vascular invasion, and nerve invasion were associated with GC recurrence or metastasis by univariate analysis (Table S2). The presence of FCCs did not show any significant association with the occurrence of recurrence and metastasis. However, the PFS time of patients who were positive for FCCs was significantly lower than that of negative patients, among patients with stage IIIC–IV disease (Fig. 3, $P = 0.0005$). The information about the two stage IV cases is in the Table S3.

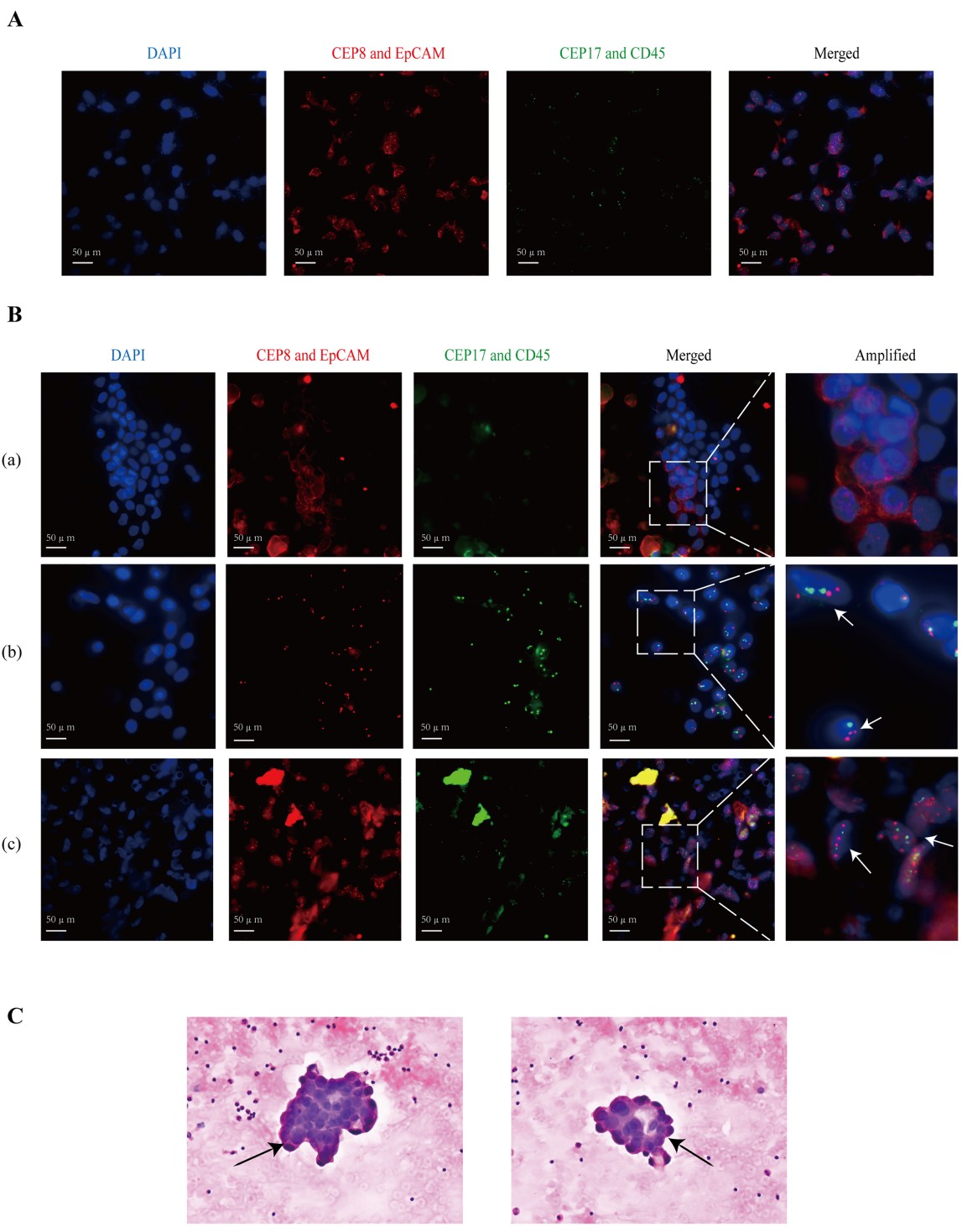

**Figure 2 The images of the positive patients.** (A) The fluorescence results of gastric cancer cells are used as a positive control. (B) Patient (a) had EpCAM (+), but CEP8 (−) and CEP17 (−) in the PLF. EpCAM is expressed in the cell membrane. CEP8 and CEP17 are considered amplified when there are at least three signals in a tumor cell nucleus. Patient (b) had CEP8 and CEP17 aneuploids and low EpCAM protein. Patient (c) showed abnormal epithelial cells, expressing the EpCAM protein, and a large number of clustered CEP8 and CEP17 aneuploids. (C) The arrows indicate cancer cells detected under a light microscope. In addition, they belong to cell cluster formation in atypical cytology. CEP8/17, chromosome ploidy 8/17; DAPI, 4, 6-diamidino-2-phenylindole; EpCAM, epithelial cell adhesion molecule; PLF, peritoneal lavage fluid.

**Table 1 The relationships between clinicopathological factors and FCCs.**

| Characteristics | Number (total = 50) | Peritoneal free cancer cells | | Univariate analysis | |
|---|---|---|---|---|---|
| | | Negative | Positive | $\chi^2$ | P value |
| **Age (y)** | | | | | |
| <68 | 22 | 20 | 2 | 4.788 | 0.029 |
| ≥68 | 28 | 18 | 10 | | |
| **Sex** | | | | | |
| Men | 39 | 34 | 5 | 9.521 | 0.002 |
| Women | 11 | 4 | 7 | | |
| **Depth of invasion** | | | | | |
| T1 | 6 | 6 | 0 | 3.570 | 0.168 |
| T3 | 8 | 6 | 2 | | |
| T4 | 36 | 26 | 10 | | |
| **Lymphatic invasion** | | | | | |
| N0 | 18 | 18 | 0 | 13.783 | 0.001 |
| N1 | 15 | 8 | 7 | | |
| N2-3 | 17 | 12 | 5 | | |
| **pTNM stage** | | | | | |
| I-II | 18 | 18 | 0 | 13.535 | 0.001 |
| IIIA-IIIB | 21 | 12 | 9 | | |
| IIIC-IV | 11 | 8 | 3 | | |
| **Venous invasion** | | | | | |
| No | 21 | 16 | 5 | 0.001 | 0.979 |
| Yes | 29 | 22 | 7 | | |
| **Nerve invasion** | | | | | |
| No | 23 | 16 | 7 | 0.967 | 0.325 |
| Yes | 27 | 22 | 5 | | |

Note:
FCCs, free cancer cells depth of invasion, lymphatic invasion and pTNM stage is based on the tumor-node-metastasis (TNM) classification on cancer staging (8th edition), jointly developed by the American Joint Commission on Cancer (AJCC) and the Union for International Cancer Control (UICC).

Using multivariate analysis, age ($P = 0.045$, HR = 0.068, 95% CI [0.005–0.947]), pTNM stage ($P = 0.020$, HR = 28.324, 95% CI [1.675–478.853]), nerve invasion ($P = 0.017$, HR = 14.161, 95% CI [1.600–125.367]), and FCCs ($P = 0.023$, HR = 14.399, 95% CI [1.445–143.522]) were risk factors for recurrence and metastasis in patients with GC (Table 2). Interestingly, we observed discrepancies between the 1-year model and the 3-year model, while the parameters of the multivariable regression analysis at 2 years remained consistent with the 3-year model (Table S4). After adjusting for covariates, patients with positive FCCs had lower PFS than patients with negative FCCs (Fig. S1). Compared with the results of microscopy, the predictive efficiency of the model (AUC = 0.688) for recurrence and metastasis was found to be better than that of traditional cytology (AUC = 0.538) (Fig. S2). The sensitivity and specificity of the prediction model constructed in this study were 84.6% and 61.1%, respectively.

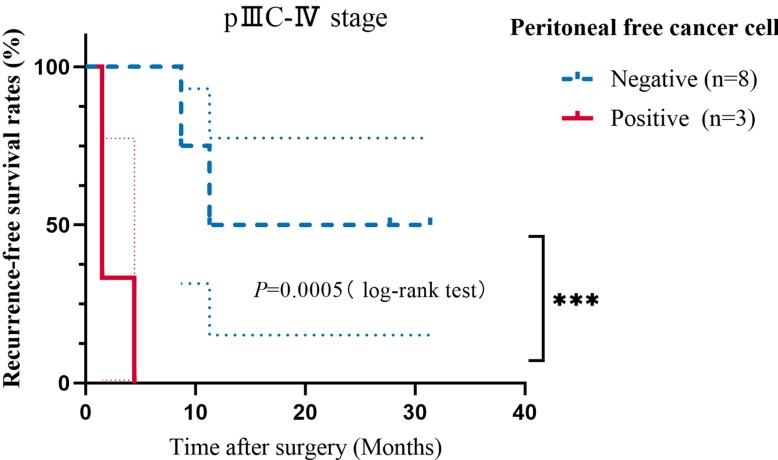

**Figure 3 Kaplan–Meier curves showing the progression-free survival analysis of patients with stage IIIC–IV GC.** Taking recurrence and metastasis as the follow-up endpoints, the patients were classified into negative and positive groups. The *P* value is shown in the figure (***$P < 0.001$).

**Table 2 Multivariate analysis of recurrence and metastasis of gastric cancer.**

| Characteristics | β | SE | Wald | P | HR | 95% CI | |
|---|---|---|---|---|---|---|---|
| Age (y) | −2.692 | 1.345 | 4.002 | 0.045 | 0.068 | 0.005 | 0.947 |
| Sex | −0.952 | 1.459 | 0.426 | 0.514 | 0.386 | 0.022 | 6.730 |
| Lymphatic invasion | −0.268 | 0.995 | 0.073 | 0.787 | 0.765 | 0.109 | 5.374 |
| pTNM stage | 3.344 | 1.443 | 5.371 | 0.020 | 28.324 | 1.675 | 478.853 |
| Nerve invasion | 2.650 | 1.113 | 5.675 | 0.017 | 14.161 | 1.600 | 125.367 |
| Peritoneal free cancer cells | 2.667 | 1.173 | 5.169 | 0.023 | 14.399 | 1.445 | 143.522 |

**Note:**
SE, standard error; CI, confidence interval; HR, hazard ratio The included variables were considered based on two aspects. Using univariate analysis of FCCs and clinicopathological characteristics, sex and age were also found to be related to FCCs. Therefore, using multivariate analysis, they were added to adjust for their confounding effect. On the other hand, the variables related to prognosis were included. Due to the problem of sample size, since the 95% CI value of vascular invasion was extremely large, it was eliminated.

## DISCUSSION

This study aimed to use PLF cytology, based on ISET and molecular biology, to increase the detection rate of FCCs and identify more OPM for subsequent early intervention. Meantime, its clinical significance on PM and prognosis was explored.

Tumor cells are heterogeneous, and markers alone are not sufficient for diagnosis. Compared with traditional cytology, our detection rate was higher (24.0% *vs.* 2.0%). Additionally, we achieved a similarly high detection rate in patients with confirmed PM (33.3% *vs.* 11.1%). This indicates that our method can effectively identify more individuals at high risk of PM. In this study, we observed that GC cells *in situ* had CEP8 and CEP17 aneuploidy, as well as a high level of EpCAM expression. Since the epithelial-mesenchymal transition is a critical component of tumor invasion and metastasis (*Krebs et al., 2012*; *Christiansen & Rajasekaran, 2006*), the shed GC cells may lose epithelial markers, which is why the positive expression of EpCAM is not high in PLF FCCs (6%). Methods solely

reliant on EpCAM overlook EpCAM-negative cells (*Gao et al., 2021*). Therefore, CEP8 and 17 polyploidies make up for this deficiency, acting as additional detection indicators, to increase cancer cell detection. Undeniably, IF-FISH demonstrates superior sensitivity and specificity compared to the Papanicolaou and Wright-Giemsa staining methods (*Chrzanowska, Kowalewski & Lewandowska, 2020*), but its implementation adds about $20 to the overall costs.

Different from other studies on technology to isolate and detect FCCs, the included patients of our study were followed up, and exploratory statistical analysis was conducted between the detection of FCCs and their prognosis, to test the predictive performance of the method. Univariate analysis showed that sex was associated with positive FCCs, suggesting a higher risk of OPM for females or reflecting a sample size limitation. In our study, the proportion of peritoneal metastasis in patients with recurrence and metastasis is as high as approximately 70%. Few events occurred due to the short duration of the study period, Kaplan-Meier curve analysis could not be performed on patients with stage IIIA–IIIB disease, so stage IIIC–IV patients were analyzed separately, showing that FCCs-positive patients had a shorter PFS than the FCCs-negative group. By univariate and multivariate analysis, pTNM stage is an independent risk factor for recurrence and metastasis in patients with GC. The univariate analysis revealed no statistically significant differences in FCCs, whereas the multivariate analysis indicated a significant association. This implies that the impact of FCCs on prognosis may be overshadowed by other factors, highlighting its greater clinical relevance in the prognosis of advanced patients. Patients with positive FCCs, particularly in stage IIIC–IV, have an increased susceptibility to PM. This technique can classify patients with GC at diagnosis, indicating a poorer prognosis for those with positive cytology in PLF. Therefore, it is advisable to implement preemptive interventional measures such as hyperthermic intraperitoneal chemotherapy (HIPEC) for this high-risk population (*Lei et al., 2020*; *Jain & Badgwell, 2023*; *Buckarma et al., 2024*; *Xu & Wang, 2023*). Our follow-up research plan is to treat FCCs-positive patients with HIPEC (using nab-paclitaxel) and compare their outcomes to those of FCCs-positive patients not receiving HIPEC and FCCs-negative patients, to explore the preventive effect of HIPEC on patients with OPM.

Due to the brief 3-year follow-up period, we conducted sensitivity analyses at one and 2 years. In the 1-year model, none of the factors reached statistical significance ($P > 0.05$), suggesting their limited impact within this timeframe. The consistency between the 2- and 3-year models suggests the robustness of our findings during this period, although other factors may also influence the PFS of GC patients. However, further external validation or validation using an independent dataset is still essential to confirm the robustness and reliability of the model.

OPM is a subclinical stage of PM that is often distinguished by the lack of PM but positive cytology (P0CY1). After adequate intervention and therapy, the incidence of PM can be lowered for patients with OPM, extending their survival time. Lacking a gold standard, the main clinical diagnostic method of OPM is laparoscopy combined with cytology. In this study, compared with traditional cytology, the sensitivity and specificity of the new method were 100% (1/1) and 77.6% (38/49) respectively, showing that it has high

accuracy. The higher detection rate suggested that this method compensates for the low sensitivity of traditional cytology. The robust correlation between detection outcomes and prognosis highlights the clinical applicability of this method. To further verify the reliability of the method, cross-validation and external validation, as well as repeatability and consistency analyses, are required.

During GC resection, surgeons routinely perform peritoneal lavage with large amounts of sterile saline. The ISET technology only comprises filtration of these lavage fluids and can easily and quickly separate suspicious free epithelial cells, facilitating subsequent identification and diagnosis of FCCs. Avoiding additional unnecessary trauma, this method can be better accepted by patients and widely promoted in hospitals, which has significant clinical translation.

However, it is essential to acknowledge the various limitations inherent in this study. To verify the recovery rate, other scholars have used a mixture of cancer and non-cancer cells to simulate ascites or PLF, which was not available for this study. In addition, since no suspicious nodules were found in all patients, peritoneal tissue biopsy was not performed, which may limit the accuracy of PM detection in some patients. This also resulted in our inability to calculate the sensitivity and specificity of this method. We plan to include peritoneal tissue biopsy-positive patients as well as noncancerous patients with ascites in a follow-up study to further validate the accuracy of the new method. Besides, in future research, we will add ISET+microscopy or FISH+EpCAM/without ISET as controls to distinctly clarify the contribution of ISET. Moreover, small sample size, short follow-up period and the single-center study limited the reliability of the multivariate analysis model. Patients with locally advanced gastric cancer (LAGC), especially those with a clinical stage of T3N+ or T4N0/+, have a high probability of PM (*Wong & Coit, 2012*). Therefore, recruiting patients with a clinical stage of T3N+ or T4N0/+ is an essential consideration in future studies.

Considering the above limitations, conducting prospective, multicenter, large-sample, and randomized controlled clinical trials is crucial for promoting and evaluating the effectiveness of the ISET technique used in this study. Our novel approach holds the potential to enhance screening of patients with GC at risk of PM to provide an opportunity for early intervention. This may reduce the incidence of PM and increase 5-year survival rate.

## CONCLUSIONS

Using the ISET+IF+FISH technique, we identified more FCCs in the PLF, which alerted clinicians to patients with a high risk of PM. This study provides an experimental basis for applying the ISET+IF+FISH technique for PLF cytology examination, as a potential detection method for patients with OPM. As a single-center study, conducting prospective, multicenter, large-sample, and randomized controlled clinical trials is imperative.

## ACKNOWLEDGEMENTS

We thank the members of Wuxi Shenrui Bio-Pharmaceuticals Co., Ltd. (Chaojie Zhang and Lushuai Yao) for their technical support.

### Funding
This work was supported by the Top Wuxi Health Committee Program (ZM004 and M202238). The funders had no role in study design, data collection and analysis, decision to publish, or preparation of the manuscript.

### Grant Disclosures
The following grant information was disclosed by the authors:
Top Wuxi Health Committee Program: ZM004 and M202238.

### Competing Interests
The authors declare that they have no competing interests.

### Author Contributions
- Ying Xuan analyzed the data, authored or reviewed drafts of the article, and approved the final draft.
- Qizhong Gao performed the experiments, analyzed the data, prepared figures and/or tables, and approved the final draft.
- Chenhu Wang performed the experiments, analyzed the data, prepared figures and/or tables, and approved the final draft.
- Dongyan Cai conceived and designed the experiments, authored or reviewed drafts of the article, project administration, funding acquisition, and approved the final draft.

### Human Ethics
The following information was supplied relating to ethical approvals (*i.e.*, approving body and any reference numbers):
The Ethics Committee of the Affiliated Hospital of Jiangnan University (approval number: LS2020060 on 20 December 2020).

### Data Availability
The raw data is available in the Supplemental File.

### Supplemental Information
Supplemental information for this article can be found online at http://dx.doi.org/10.7717/peerj.17602#supplemental-information.

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
