# Peer review of "Positive peritoneal lavage fluid cytology based on isolation by size of epithelial tumor cells indicates a high risk of peritoneal metastasis"

_PeerJ, doi:10.7717/peerj.17602_

## Round 0.1 · original submission · Major Revisions

Dear authors, thank you for your submission. Please, refer to the reviewers' comments for further details. Also note, that some of the raised concerns require a response not only to the reviewer but to include such arguments or solutions you may propose in your manuscript.

·

Basic reporting

The manuscript demonstrates professional writing and language overall, with a logical flow in the introduction and background supported by well-referenced literature. Figures and tables are effectively presented, accompanied by supplied raw data. However, some sentences need revision for accuracy.

Minor comments:
1. The manuscript examines the relationship between FCCs and recurrence/metastasis/PFS. However, peritoneal metastasis was not specifically assessed as an outcome. Therefore, phrases in the manuscript like "high risk of PM" and "prone to PM" should be reviewed for accuracy.
2. Introduction (line 54-55): The gold standard for PM diagnosis is histopathological examination of peritoneal tissue biopsy rather than ascites cytology.
3. Results (Line 167): It is unclear whether the sex ratio represents females versus males or males vs females.
4. Results (Line 201-204): When discussing the results of Cox regression, it is conventional to express them as HR and 95% CI.
5. Discussion (Line 262-266): The first sentence is not easily understandable. The significance of this test is overstated, and the advantage of ISET should be addressed.
6. Table 1: Suggest adding another column to show the data of all included patients.
7. Figure1 and S1: suggest clarify the legend "negative vs. positive." by adding "peritoneal free cancer cells".

Experimental design

The manuscript presents original medical research within the journal's scope. While the research question is well-defined and meaningful, there are concerns regarding the research design and analysis. The methods are detailed but may not fully address the research question or lead to a solid conclusion.

Major comments:
1. The research question involves applying ISET technology in peritoneal cytology analysis. However, the article tests the ISET+FISH/immunofluorescence method against traditional cytological evaluation. Since FISH is reported to improve the detection rate of FCCs over traditional cytology (doi: 10.21147/j.issn.1000-9604.2019.06.10), the contribution of ISET is unclear. It is suggested to include ISET+microscopy or FISH+EpCAM/without ISET as controls to clarify the improvement in detection rate under ISET.
2. Regarding data analysis, it is better to compare the predictive value of recurrence/metastasis using different methods through accuracy analysis, such as ROC curves.
3. The manuscript employs univariate/multivariate analysis to investigate the relationship between FCCs and prognosis. However, traditional cytology-detected FCCs are reported to have prognostic value (doi: 10.1111/cas.13219). Additionally, the limited sample size and outcome events in this study greatly limit the reliability of the multivariate analysis model. It is a general rule in regression analyses that 10 events of the outcome of interest are required for each variable in the model. Furthermore, clarity is needed on why certain variables were selected for the regression analysis, such as the inclusion of sex but not venous invasion.

Minor comments:
1. Methods (lines 111-114): would like to know more details about the cytological examination, such as criteria for identifying FCCs and whether they were reviewed by the same pathologists.
2. Methods (lines 140-142): It could be better to give a specific definition of "high" EpCAM expression. Additionally, although the results show the definition of positive FCCs, it should be outlined in the methods as well for clarity. For instance: cells exhibiting CD45-CEP8/17+EpCAM+, CD45-CEP8/17-EpCAM+, and CD45-CEP8/17+EpCAM- were considered as cancer cells.

Validity of the findings

The article explores a meaningful and interesting topic, exploring ISET technology in peritoneal fluid analysis for the first time. However, there are flaws in the experimental design and analysis, as mentioned previously. It is suggested to add control groups, adjust the multivariate analysis model, and include accuracy analysis to compare evaluation methods for a more robust answer to the original research question.

Additional comments

1. To provide a clearer overview of included patients, it is suggested to add more information about the two stage IV cases included, recurrence and metastasis details (including sites and ratio), differentiation and histological subtype (if available), neoadjuvant/perioperative therapy, and the type of surgery received (R0/R1/R2 resection).
2. Results (line 172-177): The description seems redundant as the evaluation criteria were already provided in the methods. It is suggested to provide the number and percentage of patients categorized as EpCAM+CEP8/17-, EpCAM-CEP8/17+, and EpCAM+CEP8/17+.
3. It is mentioned in the methods that suspicious lesions were pathologically biopsied. Are there any results available on that?

·

Basic reporting

Correct

Experimental design

In methods, it is commented that cytological examination was used for identifying malignant cells by means of malignant morphology.
Coud the authors explain how these cells was recognized and which criteria were used? Would be interesting to add to the paper some images of malignant cells.

Validity of the findings

Correct

Additional comments

In this work the authors to describe a method for isolate epithelial tumor cells indicating a high risk of peritoneal metastasis through its size. This matter seems interesting for readers, and I recommend publishing it.

---

## Round 0.2 · Major Revisions

Dear authors, some issues remain or have emerged that i consider still concerning. Please, check the reviewers' comments and address them. I also request you to use track changes, incl. in figures/data presentation and to include relevant "answers to reviewers' in the manuscript itself so that it improves impact but also readability and clarification for other readers. Do not forget to carefully proofread. And avoid wrong use of medical terms...

·

Basic reporting

The authors have answered to the requirements from the reviewer, so this manuscript is aceptable por publication.
Thanks.

Experimental design

N/A

Validity of the findings

Morphological description of cancer cells has been done, including two images of them.

Additional comments

N/A

Reviewer 3 ·

Basic reporting

The authors introduced a workflow to detect free cancer cells from surgical peritoneal lavage in gastric cancer patients, which first enriched tumor cells by size-based membrane filtraion (ISET), then stained cells with biomarkers (CEP8, CEP17, EpCAM) via IF-FISH. This workflow showed higher detection rate than light microscopy morphology in a cohort of 50 patients. Authors also associated clinical factors and survival related to peritoneal free cancer cell presence.

Comments
- It would be valuable to review/discuss sensitivity and specificity of other detection methods mentioned in introduction, as well as previous statistics of this method in other cancer. Likewise, you shall report sensitivity and specificity of this method, using pathology as golden criteria.
- The studied patients were not only ‘gastric cancer patients’ – they were at ‘marginal’ status of not having pre-operative peritoneal metastases but were had diseases severe enough to qualify surgery. You may want to give more precise description / naming of these individuals in the manuscript and in discussion (eg, in which patient group is cancer cell peritoneal lavage detection clinically meaningful, how, and how much).
- The 3 year follow up is relatively short. A sensitivity analysis of 1 year and 2 year can be added.
- It would be meaningful to compare prognosis, based on positivity per 1) light microscopy (negative control), 2) pathology (golden standard), and 3) this method, to evaluate how good this method is.
- You could show survival of all patients (of each stage), not only stage III-IV group. Coefficient of multi-variant regression model are important – you can consider moving it to main data.
- “Compared with the results of microscopy, the predictive efficiency of the model (AUC=0.688) was found to be better than that of traditional cytology (AUC=0.538) (Fig. S2).” – what does ‘predictive efficiency’ refer to?
- You can also discuss selection criteria for IF-FISH biomarkers, and IF-FISH against other staining methods, in sensitivity, specificity, and cost-effectiveness.

Experimental design

Please see above

Validity of the findings

Please see above

Additional comments

Please see above

·

Basic reporting

This is a pioneering trial of using isolation by size of epithelial tumor cells (ISET) method in the detection of gastric cancer cells in the peritoneal fluid. The results showed that the discovery rate of the malignant cells outperformed the traditional cytology and is correlated to worse PFS in late-stage patients. The statistics in this paper have undergone careful revision with the help of the previous reviewer, however, some concerns about concepts and details need to be addressed before publication.

Experimental design

Major comments:
1. This manuscript used the terms ‘peritoneal fluid’ and ‘ascites’ interchangeably. Ascites is a medical condition in which fluid collects in spaces within a patient’s abdomen. In contrast, peritoneal fluid is created by lavage procedure during laparoscopic exploration. The author should clarify they studied cytology in peritoneal lavage fluid but not ascites (e.g. lines 1-2) in the whole passage.
2. Is this method aimed at improving early diagnosis or prognosis prediction? If the ISET technique is aimed to be an early diagnosis tool (line 19,35,297), then it should be extracted by paracentesis in early-stage patients with suspicion of gastric cancer, but not by laparoscopic surgery with 250ml of peritoneal lavage fluid. I assume it is more appropriate to say this technique will likely be used to classify patients at diagnosis, which aligns with the findings that positive cytology in peritoneal fluids was correlated with worse PFS.
3. The study tried to compare the ISET method versus traditional cytology (line 192, 226). First, the figure 1C can’t support this claim. Second, the patient subgroup should be those with validated peritoneal metastasis but not all that had gastric cancer.

Validity of the findings

1. In figure 2, the patient number in each subgroup should be demonstrated.
2. In figure 2, ‘Surgery’ is misspelled as ‘Surgey’.
3. In table 1, why was the age cutoff 68? Is it according to any guidelines or cohort study about elder patients with gastric cancer? Generally, 65 is the cutoff age for older adults.
4. HIPEC was not performed in this study. It is allowed to appear in the discussion but not in the results/conclusions (line 306).

Additional comments

Minor comments:
1. ‘China reporting 35.9% (2010–2014)’ (line 40). This data was published ten years ago and should be updated.
2. “Head-high-foot-low posture” (line 111) is referred to as “Reverse Trendelenburg Position” in English.
3. The discussion section compared this ISET study with the other investigations that combine microfluid and single-cell sequencing methods (lines 244-250). The authors are advised to detail this comparison technically by referring to method papers, including but not limited to these:
Reference: Ann Transl Med. 2019 Dec;7(23):790. doi: 10.21037/atm.2019.11.116. ; Am J Pathol. 2000 Jan;156(1):57-63. doi: 10.1016/S0002-9440(10)64706-2.

---

## Round 0.3 · Major Revisions

Dear authors, thank you for your resubmission. This was a difficult decision because we still have doubts about your design, in particular.

Specifically, it is a method evaluation lacking a definition or quantification of true positives (through peritoneal membrane biopsy) and does not include a negative control. This absence means it is not possible to identify true negatives, which is a critical aspect of method evaluation.

While your aim to demonstrate that one new method has increased sensitivity over another is clear, and promoting this new method is beneficial, the deduction workflows do not meet the standard criteria for method evaluation. Consequently, your study is more aligned with the evidence level of a case series rather than a robust evaluation.

Given our significant progress thus far, I strongly suggest that you address the reviewers' comments comprehensively in your manuscript. Please undertake a thorough revision, clearly outlining the different experiments and acknowledging the potential limitations, particularly in your workflow. You should defend your study design, but this defense must be reflected in a well-articulated discussion within your manuscript.

By addressing these issues, I hope, you will enhance the rigor and reliability of your study, thereby increasing confidence in its potential contribution to the scientific literature. The "major revisions" is merely representative of the amount of effort that I believe you should put in into addressing this problem (that has been highlighted from the start); not a representation of lower quality of your work,.. far from it!

Reviewer 3 ·

Basic reporting

Reviewer 3
The authors introduced a workflow to detect free cancer cells from surgical peritoneal lavage in gastric cancer patients, which first enriched tumor cells by size-based membrane filtraion (ISET), then stained cells with biomarkers (CEP8, CEP17, EpCAM) via IF-FISH. This workflow showed higher detection rate than light microscopy morphology in a cohort of 50 patients. Authors also associated clinical factors and survival related to peritoneal free cancer cell presence.

Comments
1. It would be valuable to review/discuss sensitivity and specificity of other detection methods mentioned in introduction, as well as previous statistics of this method in other cancer. Likewise, you shall report sensitivity and specificity of this method, using pathology as golden criteria.
Addressed within current capability.
In an optimal scenario, sensitivity and specificity of this method should be reported. However here all patients are true positive (correct?) thus specificity (inclusion of non-cancer patients as control) are unable to obtain.

2. The studied patients were not only ‘gastric cancer patients’ – they were at ‘marginal’ status of not having pre-operative peritoneal metastases but were had diseases severe enough to qualify surgery. You may want to give more precise description / naming of these individuals in the manuscript and in dicussion (eg, in which patient group is cancer cell peritoneal lavage detection clinically meaningful, how, and how much).
Addressed.
Though the content of ‘subsequent auxiliary and intensive treatment’ can be more specified with examples and proper citation of currrent guidelines.

3. The 3 year follow up is relatively short. A sensitivity analysis of 1 year and 2 year can be added.
Partially addressed.
However, please also add HR and 95% CI (not only p value) in Table S4.

4. It would be meaningful to compare prognosis, based on positivity per 1) light microscopy (negative control), 2) pathology (golden standard), and 3) this method, to evaluate how good this method is.
Partially addressed.
By ‘gold standard’ I had meant how the ‘validated peritoneal metastasis’ was defined as in this manuscript. So was peritoneal tissue biopsy conducted in this study, or alternative criteria was used to define the true positive?
I think this comment goes similarly with 2.

4. You could show survival of all patients (of each stage), not only stage III-IV group. Coefficient of multivariant regression model are important – you can consider moving it to main data.
Addressed

5. “Compared with the results of microscopy, the predictive efficiency of the model (AUC=0.688) was found to be better than that of traditional cytology (AUC=0.538) (Fig. S2).” – what does ‘predictive efficiency’ refer to?
Addressed

6. You can also discuss selection criteria for IF-FISH biomarkers, and IF-FISH against other staining methods, in sensitivity, specificity, and cost-effectiveness.
Addressed

Experimental design

Please see above

Validity of the findings

Please see above

Additional comments

Please see above

·

Basic reporting

The authors have addressed most of my concerns in this revision. It is ready to be published in your journal after language improvement including the my last suggestion in 2.Experiment design.

Experimental design

2. The author has clarified the aim of their method is to help early prevention of peritoneal metastases. To make the writing more clear, I suggest them to write "..opportunity for early intervention of PM and an extension of patient survival" (line 35), and "Our novel approach holds the potential to enhance screening and early intervention for patients with GC at risk of PM"(line 340-342). These may help you to convey your ideas concisely.

Validity of the findings

All of my questions are appropriately answered.

Additional comments

All of my questions are appropriately answered.

---

## Round 0.4 · accepted · Accept

Dear authors, I am happy to let you know that your manuscript is now acceptable for publication. Many thanks for your hard work and extensive efforts.

Reviewer 3 ·

Basic reporting

Comments
1/4. Okay with authors’ statement of limitation.
2. Addressed
3. Addressed

Experimental design

/

Validity of the findings

/

Additional comments

/